# Environmental Exposure to Persistent Organic Pollutants and Its Association with Endometriosis Risk: Implications in the Epithelial–Mesenchymal Transition Process

**DOI:** 10.3390/ijms25084420

**Published:** 2024-04-17

**Authors:** Ana Martín-Leyva, Francisco M. Peinado, Olga Ocón-Hernández, Alicia Olivas-Martínez, Antonio Luque, Josefa León, Inmaculada Lendínez, Jesús Cardona, Ana Lara-Ramos, Nicolás Olea, Mariana F. Fernández, Francisco Artacho-Cordón

**Affiliations:** 1Radiology and Physical Medicine Department, University of Granada, E-18016 Granada, Spain; amleyva@correo.ugr.es (A.M.-L.); nolea@ugr.es (N.O.); marieta@ugr.es (M.F.F.); 2Biohealth Research Institute in Granada (ibs.GRANADA), E-18012 Granada, Spain; franciscopeinado@correo.ugr.es (F.M.P.); ooconh@ugr.es (O.O.-H.); aolivas@ugr.es (A.O.-M.); anluque@ugr.es (A.L.); pepileon@ugr.es (J.L.); 3Centre for Biomedical Research, University of Granada, E-18016 Granada, Spain; 4Gynaecology and Obstetrics Unit, ‘San Cecilio’ University Hospital, E-18016 Granada, Spain; jescarcon@gmail.com; 5Digestive Medicine Unit, ‘San Cecilio’ University Hospital, E-18012 Granada, Spain; 6CIBER Hepatic and Digestive Diseases (CIBEREHD), E-28029 Madrid, Spain; 7General Surgery, San Cecilio University Hospital, E-18016 Granada, Spain; inmalendinez10@gmail.com; 8Gynaecology and Obstetrics Unit, ‘Virgen de las Nieves’ University Hospital, E-18014 Granada, Spain; ana_lara_ramos@hotmail.com; 9CIBER Epidemiology and Public Health (CIBERESP), E-28029 Madrid, Spain; 10Nuclear Medicine Unit, ‘San Cecilio’ University Hospital, E-18016 Granada, Spain

**Keywords:** endometriosis, persistent organic pollutants, organochlorine pesticides, polychlorinated biphenyls, epithelial–mesenchymal transition

## Abstract

We aimed to explore the relationship of adipose tissue concentrations of some persistent organic pollutants (POPs) with the risk of endometriosis and the endometriotic tissue expression profile of genes related to the endometriosis-related epithelial–mesenchymal transition (EMT) process. This case–control study enrolled 109 women (34 cases and 75 controls) between January 2018 and March 2020. Adipose tissue samples and endometriotic tissues were intraoperatively collected to determine concentrations of nine POPs and the gene expression profiles of 36 EMT-related genes, respectively. Associations of POPs with endometriosis risk were explored with multivariate logistic regression, while the relationship between exposure and gene expression profiles was assessed through Spearman correlation or Mann–Whitney U tests. After adjustment, increased endometriosis risk was associated with *p,p’*-DDT, PCB-180, and ΣPCBs. POP exposure was also associated with reduced gene expression levels of the CLDN7 epithelial marker and increased levels of the ITGB2 mesenchymal marker and a variety of EMT promoters (HMGA1, HOXA10, FOXM1, DKK1, CCR1, TNFRSF1B, RRM2, ANG, ANGPT1, and ESR1). Our findings indicate that exposure to POPs may increase the risk of endometriosis and might have a role in the endometriosis-related EMT development, contributing to the disease onset and progression. Further studies are warranted to corroborate these findings.

## 1. Introduction

Endometriosis is among the most prevalent disorders in women of childbearing age, with a suspected worldwide prevalence of 10–15% of women of childbearing age [1,2,3]. This hormone-dependent disease is defined by the presence of endometrial-like tissue outside the uterine cavity [4]. Moreover, the peritoneal microenvironment surrounding lesions also plays a crucial role in the pathophysiology of this condition [5]. In addition to menstrual irregularities and fertility issues, chronic pelvic pain stands out as the predominant symptom, often worsening during menstruation (dysmenorrhea), sexual activity (dyspareunia), bowel movements (dyschezia), or urination (dysuria) [6,7]. Furthermore, it has been observed that endometriosis correlates with an increased risk for autoimmune disorders, fibromyalgia, chronic fatigue syndrome, gynecological cancers, and adenomyosis in affected women [8,9,10,11].

The precise elucidation of the origins and pathophysiology of endometriosis remains elusive. However, the disease is widely acknowledged to possess a multifactorial nature, wherein hormonal, genetic, lifestyle, and environmental factors collectively contribute to the disease risk [4]. In this regard, it has been pointed out that human daily exposure to the so-called endocrine-disrupting chemicals (EDCs) might be related to an increased risk for endometriosis [12]. They are a wide group of synthetic chemicals that can alter the homeostasis of the endocrine system and are associated with multiple adverse effects [13]. Among EDCs, persistent organic pollutants (POPs) constitute a heterogeneous group of synthetic organic compounds used worldwide until the 1980s for pest control, sanitation, and agricultural and industrial purposes. They include organochlorine pesticides (OCPs), such as dichlorodiphenyltrichloroethane (DDT) and its metabolite *p,p’*-dichlorodiphenyldichloroethylene (*p,p’*-DDE), hexachlorobenzene (HCB), and hexachlorohexane (HCH), and polychlorinated biphenyls (PCBs) congeners. Although most countries banned or severely restricted the use of POPs, considerable research efforts have raised awareness about the ongoing and continuous exposure of the general population, attributed to the high persistence and lipophilia of these compounds and their consequent bioaccumulation and biomagnification in the food chain [14]. Adipose tissue is, therefore, considered the main reservoir of these compounds, accounting for all routes and sources of exposure, and represents a stable and long-term biomarker of exposure to these chemicals [15,16,17]. The relationship between POP exposure and increased risk for this estrogen-dependent disease is biologically plausible since many OCPs and PCBs have been shown to interact with estrogen and/or androgen receptors [18,19,20,21,22], and epidemiological evidence has been published for a variety of hormone-dependent diseases, e.g., breast cancer [23,24], anovulation [25], ovarian function [26,27], infertility [28], preterm birth [29], polycystic ovarian syndrome [30], fibroids [31], and endometriosis [32]. Nevertheless, while some authors pointed out that some POPs might be related to an increased risk for endometriosis [33,34,35], others have not found any association between POP exposure and endometriosis risk [36,37,38].

Moreover, it has been suggested that EDCs may have a crucial impact not only on the onset but also on the progression of endometriosis. Of particular interest, epithelial-to-mesenchymal transition (EMT) has been described as a crucial process during the development and progression of endometriosis [39]. Specifically, it has been suggested that Type 2 (that occurs in response to wound or inflammatory response, leading to tissue fibrosis) and Type 3 EMT (that occurs in epithelial cancer cells that differ genetically and epigenetically from untransformed epithelial cells and favor clonal outgrowth and the development of localized tumors) might be involved in this disease [40]. As schematized in Appendix A, EMT is a physiological process in which cells lose the epithelial features such as E-cadherin, desmoplakin, occluding, and claudins and gain properties of mesenchymal cells such as N-cadherin, vimentin, and some integrin heterodimers. The matrix metalloproteinase (MMP) family also plays a crucial role in this EMT since they are involved in the breakage of cell–cell and cell–extracellular matrix unions [41,42]. All these molecular changes are associated with the alteration of cell functions such as enhanced migration, invasiveness, and resistance to apoptosis [43]. This complex process is driven by various signaling pathways, such as the TFG-β, PDGFRα, Wnt, and Notch pathways [40], some of them converging in intracellular cascades such as Smad, β-catenin, NF-κB, PI3K/AKT, or MAPK/ERK, which finally activates the key EMT regulators [Snail, Slug, Zinc finger E-box-binding homebox (ZEB1/2) or Twist] [40,43,44]. Moreover, pro-inflammatory mediators such as interleukins (ILs), tumor necrosis factor (TNF), and chemokines have also been widely reported to trigger EMT through STAT3/JAK, NF-κB, MAPK/ERK, or Smad internal cascades [45,46]. Additionally, estrogens and hypoxia-related factors [angiogenin (ANG), angiopoietin 1 (ANGPT1), or vascular endothelial growth factor (VEGF)] have been suggested to promote endometriosis-related EMT via MAPK/ERK, PI3K/AKT, or β-catenin cascades [47,48]. Many soluble factors have been described to be up- or downstream activators or inhibitors of these cellular pathways. For instance, HMGA1, FOXM1, RXRA, FUT-8, POU5F1, or HOXA10 gene products promote EMT, while DUSP6 and AGR2 gene products decreased EMT via the TGF-β signaling pathway [49,50,51,52,53,54,55,56]. Similarly, DDK1, CDK1, or SOX2 and CCR1 or CHUK have been shown to regulate Wnt-driven and Notch/NF-κB-driven EMT, respectively [57,58,59,60,61,62]. Other soluble factors, such as the RRM2 gene product, promote EMT by targeting JAK2/STAT3 intracellular cascade [63]. To date, most of the efforts made to address adverse effects associated with human exposure to POPs did not focus on the physiopathological changes underlying hormone-dependent diseases, although in vitro evidence suggests that exposure to some POPs may promote molecular biomarkers related to enhanced migration and invasiveness ability of ovarian granulosa cancer cells such as MMPs [64]. Moreover, Zucchini-Pascal et al. [65] pointed out that POPs may orchestrate EMT in human primary cultured hepatocytes, observing the repression of epithelial markers (cell–cell junctions and E-cadherin) and increased the abundance of various mesenchymal genes.

Thus, given the current discrepancies on the influence of POPs in the development of endometriosis, the present study was designed to explore the association between the bioaccumulated concentrations of OCs and PCBs and the risk of endometriosis in women of childbearing age. Finally, given the crucial role of the EMT signaling pathway in disease onset and the in vitro evidence of the link between POP exposure and EMT, the secondary aim of this study was to assess the influence of this exposure on the expression profiles of EMT-related genes in the endometriotic tissue.

## 2. Results

### 2.1. Adipose Tissue Concentrations of Selected POPs

As shown in Table 1, all studied OCPs and PCBs were detected in >85% of analyzed omental adipose tissue samples. The OCP with the highest median concentration was *p,p’*-DDE (81.09 ng/g tissue) followed by β-HCH (12.22 ng/g tissue), while the most concentrated PCB congener was PCB-180 (28.05 ng/g tissue). In comparison to controls, a significantly higher median concentration of PCB-138 (20.29 vs. 13.74 ng/g tissue), PCB-153 (26.76 vs. 18.50 ng/g tissue), PCB-180 (39.04 vs. 20.54 ng/g tissue), ∑PCBs (0.25 vs. 0.15 nmol/g tissue), and ∑POPs (0.76 vs. 0.52 nmol/g tissue) were observed in cases, while the differences did not reach the statistical significance for *p,p’*-DDT (1.72 vs. 1.40 ng/g tissue; *p*-value = 0.065), *p,p’*-DDE (118.52 vs. 74.33 ng/g tissue; *p*-value = 0.067), and ∑OCPs (0.55 vs. 0.35 nmol/g tissue; *p*-value = 0.072).

### 2.2. Associations between Chronic Adipose Tissue POP Concentrations and Endometriosis Risk

Table 2 exhibits the associations found between adipose tissue concentrations of OCPs and PCBs and the risk of endometriosis. After adjusting for age, BMI, residence, and parity, an increased risk of endometriosis was found to be significantly associated with adipose concentrations of *p,p’*-DDT (OR 1.63, 95% CI 1.13–2.34), PCB-180 (OR 1.89, 95% CI 1.09–3.27), and ∑PCBs (OR 1.78, 95% CI 1.02–3.12), while the association with PCB-138 and PCB-153 and ∑POPs did not reach the statistical significance [(OR 1.65, *p*-value = 0.086), (OR 1.65, *p*-value = 0.075), and (OR 1.75, *p*-value = 0.056), respectively]. When participants were classified by tertiles of each POP, a significantly increased risk was observed for the women in the third versus first tertile of PCB-180 (OR 8.48, 95% CI 1.88–38.13) and ∑PCBs (OR 5.33, 95% CI 1.39–20.41) (Figure 1). A similar trend but close to the statistical significance was also found for PCB-138 (OR 2.97, 95% CI 0.89–9.90; *p*-value = 0.077) and ∑POPs (OR 3.03, 95% CI 0.89–10.31; *p*-value = 0.076).

Similarly, a close-to significant increased risk was observed for the women in the second versus first tertile of PCB-180 (OR 3.36, 95% CI 0.96–11.75; *p*-value = 0.058).

Likewise, higher concentrations of almost all analyzed POPs were observed in women with stages III/IV versus stages I/II endometriosis, although they did not reach statistical significance (Appendix A).

### 2.3. Associations between Adipose Tissue Levels of POPs and Expression Profile of EMT-Related Genes

Expression profiles of most of the included genes have been reported previously [66,67], given their participation in other endometriosis-related cell signaling pathways, and are summarized in Appendix A. A total of 18 genes (50.0%) were expressed in >75% of the samples, while 14 (38.9%) were expressed in 25–75% of samples. Compared to those with I/II endometriosis, increased mean gene expression in endometriotic tissue from women with III/IV endometriosis was observed for MMP7 (0.10 vs. 0.00), FUT8 (1.14 vs. 0.02), and SOX2 (1.51 vs. 0.36). Although differences did not reach statistical significance, it was also detected higher expression of POU5F1 (3.13 vs. 1.18, *p*-value = 0.054), RXRA (3.90 vs. 1.59, *p*-value = 0.055), and JUN (1.27 vs. 0.12, *p*-value = 0.097), and lower gene expression of DUSP6 (4.97 vs. 8.10, *p*-value = 0.072).

Associations between endometriotic gene expression and adipose tissue POP concentrations are displayed in Table 3. Regarding the core EMT markers, lower CLDN7 and higher ITGB2 gene expression levels were associated with higher concentrations of *o’,p’*-DDT, *p’,p’*-DDT, *p’,p’*-DDE, ΣOCPs, and ΣPOPs. HCB concentrations were also positively correlated with ITGB2 expression levels, although they did not reach statistical significance (Spearman rho = 0.334, *p*-value = 0.072).

Among TGF-β- and PDGFRα-related EMT promoters, FOXM1 and HMGA1 gene expression levels were positively correlated with *o’,p’*-DDT, *p’,p’*-DDT, *p’,p’*-DDE, ΣOCPs, and ΣPOPs. Moreover, FOXM1 was also correlated with higher HCB concentrations, while the positive correlations found between the pairs HMGA1-HCB, HMGA1-β-HCH, and FOXM1-β-HCH did not reach statistical significance (Spearman rhos = 0.338–0.349, *p*-values = 0.059–0.067). Similarly, increased HOXA10 expression levels were positively associated with *p’,p’*-DDE, ΣOCPs, ΣPOPs, and *o’,p’*-DDT, although the latter did not reach statistical significance (Spearman rho = 0.320, *p*-value = 0.084). Moreover, an inverse close-to-significant association was observed between DUSP6 gene expression and concentrations of β-HCH and ΣOCPs (Spearman rho = −0.308, *p*-value = 0.098; Spearman rho = −0.307, *p*-value = 0.098, respectively). Likewise, detectable gene expression levels of FUT8 were associated with higher concentrations of PCB-138 and PCB-153, although they did not reach the statistical significance (*p*-values = 0.078 and 0.086, respectively), as well as PU5F1 detectable levels with β-HCH concentration (*p*-value = 0.078).

Concerning Wnt- and Notch/NFκB-related EMT inducers, positive associations were found between DKK1 and CCR1 gene expression levels and concentrations of *o’,p’*-DDT, *p’,p’*-DDT, *p’,p’*-DDE, ΣOCPs, and ΣPOPs. A positive close-to-significant association was also observed between CCR1 expression levels and β-HCH concentrations (Spearman rho = 0.335, *p*-value = 0.070).

Regarding inflammation-related EMT enhancers, RRM2 expression levels were associated with higher concentrations of *o’,p’*-DDT, *p’,p’*-DDT, *p’,p’*-DDE, HCB, ΣOCPs, and ΣPOPs, while the association with β-HCH did not reach the statistical significance (Spearman rho = 0.317, *p*-value = 0.088). Moreover, detectable gene expression levels of TNFRSF1B were related to higher concentrations of *p’,p’*-DDT, while the associations found for *o’,p’*-DDT, *p’,p’*-DDE, and ΣOCPs did not reach the statistical significance (*p*-values = 0.081, 0.081, and 0.067, respectively).

Regarding estrogen- and growth factor-related EMT modulators, increased ESR1 gene expression was associated with higher concentrations of *o’,p’*-DDT and *p’,p’*-DDE, while the associations found with *p’,p’*-DDT, ΣOCPs, and ΣPOPs did not reach the statistical significance (*p*-values = 0.075, 0.060, and 0.078, respectively). Finally, with regard to the hypoxia-related EMT promoters, positive associations between ANG gene expression and concentrations of *o’,p’*-DDT, *p’,p’*-DDT, and *p’,p’*-DDE, while the association with ΣOCPs did not reach statistical significance (*p*-value = 0.094).

## 3. Discussion

This study reveals a relationship between bioaccumulated exposure to POPs and the risk of endometriosis in women of childbearing age, which was greater in the women with higher omental adipose tissue concentrations of *p,p’*-DDT and PCB-180, as well as ΣPCBs. Moreover, this is the first study to explore the potential association between exposure to POPs and the disruption in the expression profiles of genes involved in the endometriosis-related EMT process, revealing that exposure to certain POPs, such as *p,p’*-DDT, were associated with higher levels of several EMT-promoting genes and with reduced levels of some inhibitors of this process.

During the last decades, an effort has been made to explore the relationship between POP exposure and perturbations in the homeostasis of gynecological tissues. Hence, researchers observed that POPs significantly reduced decidualization in primary human endometrial stromal cells [68], as well as the migration and invasiveness of both endometrial epithelial and stromal cells [69,70]. Consistent with these in vitro findings, previous epidemiological studies [33,34,35,71,72,73,74,75] evidenced a positive association between endometriosis risk and exposure to certain POPs, including some PCB congeners, HCB, and β-HCH, in line with our findings revealing significant (or close-to-significant) associations with *p,p’*-DDT, PCB-138, PCB-153, ΣPCBs, and ΣPOPs, either when the exposure was considered as continuous or categorical variables, showing increasing odds for those women in the second and third tertile. Interestingly, Martínez-Zamora et al. [75] and Ploteau et al. [71] only found significant associations in women with severe endometriosis. In this regard, although we observed higher concentrations of most POPs in women with III/IV endometriosis, none of them reached statistical significance, probably given the reduced sample size of this subgroup analysis. Nevertheless, various studies reported null associations [36,37,38,76,77,78,79]. The divergent findings can be partially explained by the heterogeneity of populations, methodologies, and experimental designs employed across the various studies.

However, most of the previous epidemiological evidence linking endometriosis to POP exposure was limited to exploring the association with disease risk, but they did not explore whether these POPs may disrupt potential endometriosis-related adverse outcome pathways. In this regard, as mentioned above, EMT has been described as a crucial process favoring the development of endometriotic lesions and endometriosis-related fibrosis [40]. In this study, we observed that the CLDN7 epithelial marker in the endometriotic lesions was inversely associated with some POPs, while the ITGB2 mesenchymal marker was positively associated with them, suggesting that POP exposure might favor EMT in endometriotic lesions. In this regard, Zucchini-Pascal et al. [65] demonstrated that in vitro exposure to three OCPs caused the repression of epithelial markers (cell–cell junctions and E-cadherin) and increased the abundance of various mesenchymal genes, including vimentin, fibronectin, and its receptor ITGA5 in human primary cultured hepatocytes. Bratton et al. [80] also reported up-regulation of ITGA6 in breast cancer cells after exposure to DDT, another marker of mesenchymal phenotype [81]. Moreover, Hu, et al. [69] observed that exposure to PCB-104 of primary cultures of eutopic endometrial tissues from women with endometriosis increased migration and invasion through the up-regulation of mesenchymal markers such as MMP3 and MMP10 gene expression. Similarly, endometrial stromal cells exposed to HCB also enhanced migration and invasiveness ability involving ER signaling pathway [82].

Our findings also suggest that POPs may enhance some key regulators of the endometriosis-related EMT process. TGFβ/PDGFRα, WNT/β-catenin and Notch/NFκB have been traditionally described as the main signaling pathways promoting EMT [40]. Interestingly, we have found that higher concentrations of some OCPs, PCBs as well as ΣPOPs were positively correlated with the gene expression of a variety of promoters related to these signaling pathways such as PDGFRα, HMGA1, HOXA10, CCR1, FOXM1 or FUT8 genes. A similar trend was observed for DKK1, though the available evidence attributes a dual role to DKK1 during the EMT process [57,58]. These results are in line with previous studies suggesting up-regulation of PDGFRα gene expression in estrogen-dependent cells, such as epithelial mammary cells exposed to a pesticide mixture [83]. DDT has also been shown to activate the p38 MAPK pathway [84], one of the main intracellular cascades driven by TGFβ/PDGFRα receptors in human endometrial adenocarcinoma cells. Human hepatocyte cultures exposed to some OCPs also increased the expression of the DKK1 gene [85]. It has also been recently reported that in vivo models of endometrial cancer exposed to other endocrine-disrupting chemicals such as diethylstilbestrol (DES) aberrantly activated the WNT/β-catenin and TGFβ/PDGFRα-related PI3K/AKT signaling pathways [86].

In addition, chronic inflammation has been recently identified as a key inducer of EMT through SMAD, NFκB, and/or STAT3 intracellular cascades [87,88,89]. Moreover, we previously reviewed the ability of POPs to stimulate a pro-inflammatory milieu [90]. Interestingly, we observed increased expression levels of the TNFRSF1B gene in those women with increased concentrations of *o,p’*-DDT, *p,p’*-DDT, *p,p’*-DDE, and PCB-138, which is supported by previous evidence that also reported the up-regulation of the IL6R, TNF, and CXCL8 genes after exposure to DDT in breast cancer cells [80,91]. Human hepatocyte cultures exposed to some OCPs also up-regulated pro-inflammatory markers such as IL17RB [85]. Moreover, in line with previous studies on mouse thymoma cells exposed to tributyltin oxide, another EDC [92], we observed positive correlations between some POPs and increased expression of the RRM2 gene, which has been reported to promote EMT through the STAT3 cascade [63].

Finally, estrogen and hypoxia signals may also stimulate endometriosis-related EMT [47]. Estrogens may promote EMT via ERα, probably given its ability to directly bind to hepatocyte growth factor promoters [93]. In this regard, our findings reflect higher expression levels of the ERα gene (ESR1) in those women with higher *o,p’*-DDT and *p,p’*-DDE concentrations. In line with these findings, Grünfeld and Bonefeld-Jorgensen [20] and Bratton et al. [80] also reported an up-regulation in hormone receptors such as ESR1 and the progesterone receptor (PGR) in breast cancer cells after exposure to DDT and other OCPs. Regarding hypoxia-related genes, VEGF and other pro-angiogenic factors, such as ANG and ANGPT1, are markedly expressed in hypoxic conditions and suspected to promote EMT though the NFκB and β-catenin intracellular cascades [94]. We observed that ANG gene expression was positively correlated to concentrations of *o,p’*-DDT, *p,p’*-DDT, and *p,p’*-DDE concentrations, in line with numerous studies that reported the OCP-related up-regulation of pro-angiogenic gene mRNA levels [80,95,96].

Limitations of this study include the relatively small sample size, reducing the statistical power and the possibility of examining differences in risk between stages of endometriosis and associations with gene expression profiles in endometriotic tissues. However, it is worth mentioning that consistent statistically significant relationships were identified even after adjusting for multiple potential confounders. Additionally, given that endometrium samples were not collected, the observed correlations between exposure and EMT gene expression in endometriotic tissue were not compared to those in endometrium. Moreover, the limited amount of endometriotic tissue samples available prevented us from performing protein validation analysis to confirm the reported gene expression results. Furthermore, in this study, we did not consider the location of the analyzed endometriotic samples (endometriomas, deep infiltrating lesions…) that might show different gene expression profiles. In addition, we cannot rule out that the associations found with single chemicals are a surrogate of exposure to other unmeasured pollutants with similar physicochemical properties or even to mixtures of pollutants that exert a combined effect. In this regard, only congeners from two families of POPs were considered in this study, although humans are simultaneously exposed to a very wide range of EDCs, including other POPs such as dioxins and furans, as well as polybrominated and perfluorinated compounds. Finally, our case–control/cross-sectional design prevented us from inferring a causal association between POPs and both endometriosis risk and gene expression profiles, although adipose tissue is the preferential reservoir for POP accumulation, and concentrations in this matrix are considered indicative of long-term exposure [97]. Strengths of this study also include the laparoscopic confirmation of the presence (cases) or absence (controls) of endometriosis, ensuring that controls did not have asymptomatic or undetectable lesions by magnetic resonance imaging, as may be the case of those studies assessing exposure in serum samples. Furthermore, biological samples were collected from cases and controls under the same conditions, increasing the comparability between cases and controls with respect to previous habits. Another strength is the simultaneous measurement of a variety of POPs in omental adipose tissue and numerous genes involved in the endometriosis-related EMT process in endometriotic tissue. In this regard, it is important to mention that it is highly challenging to gather endometriotic tissue samples for investigation. Importantly, our findings support consistent associations between gene expression and exposure to certain POP congeners. Finally, a key contribution of this study is the combined examination of biomarkers of exposure, potential biomarkers of effect, and health outcomes to determine the potential impact of POP exposure on endometriosis-related adverse outcome pathways.

Our findings shed light on the potential of POPs accumulated in the omental adipose tissue to trigger the risk of endometriosis, as well as to disrupt a wide variety of genes related to the EMT process. It is necessary to clarify the underlying mechanisms involved in the development and progression of endometriosis, and therefore, the need to identify preventable risk factors and establish intervention measures to halt and potentially reverse the progression of this female disease.

## 4. Materials and Methods

### 4.1. Study Population and Sample Collection

The population of this hospital-based case–control study consisted of 109 women of childbearing age. A total of thirty-four cases and seventy-five controls were enrolled and matched by frequency during the period from January 2018 to March 2020 in two public hospitals (‘San Cecilio’ and ‘Virgen de las Nieves’) in Granada, Southern Spain. For the secondary aim of this study, a cross-sectional design was adopted, including only women from the case group.

Cases were defined as women with histological confirmation of endometriosis after a laparotomy or laparoscopic surgery, while controls were women undergoing a similar surgical procedure but for non-malign diseases (including acute appendicitis, biliary disease, hiatus hernia, ovarian torsion, corpus luteum, uterine fibroids, and cystadenomas), in whom the absence of endometrial lesions was visually and histologically confirmed. Additional criteria for all women included age between 20 and 54 years. Exclusion criteria for all participants were body mass index (BMI) > 35 kg/m^2^, previous history of cancer (except non-melanoma skin cancer), pregnancy at recruitment, and failure to provide informed consent.

In the study, healthcare professionals apprised each patient of the study’s aims and invited them to sign an informed consent form, which was approved by the Research Ethics Committee of Granada (0464-N-18).

Following the protocols outlined in the EHPect project, standardized procedures were employed for the collection and storage of biological samples, as well as for administering standardized questionnaires to gather data. Hence, approximately 100–150 mg of omental adipose tissue from cases and controls and 30–50 mg of endometriotic tissue from cases were gathered during surgery and kept in QIAazol reagent (Qiagen, Hilden, Germany) to ensure RNA stability. Samples were stored at −80 °C at the Biobank of the Public Andalusian Health Care System. EHPect surgical and clinical questionnaires were used to gather sociodemographic, lifestyle, clinical, and surgical information, including age (in years), place of residence (urban or suburban), level of education (university degree or lower), employment status (working outside the home or not), current smoking status (yes or no), parity (nulliparous, primiparous, or multiparous), and average intensity of menstrual bleeding (mild, moderate, or severe).

As shown in Table 4, no difference was found between cases and controls in terms of age, BMI, residence, educational level, working status, smoking habits, parity, or menstrual bleeding intensity (*p*-values > 0.050). Among the 34 cases, 22 (64.7%) and 12 (35.3%) were diagnosed with stage I/II and III/IV endometriosis, respectively.

### 4.2. Adipose Tissue Sample Extraction and Chemical Analyses

The extraction protocol was adapted from Vela-Soria et al. [98]. Approximately 100 mg of omental adipose tissue was weighted and enzymatically treated with 2.5 mg clostridium histolyticum collagenase (Merck, Darmstadt, Germany) dissolved in 125 μL phosphate-buffered saline (PBS) (10% 500 mM calcium chloride) in an orbital shaker at 37 °C overnight. A total of 10 μL of a solution in acetonitrile of internal standards (1 mg/L) was added, and homogenates were diluted with 4 mL hexane, manually shaken for 60 s, sonicated in an ultrasound batch for 15 min, and centrifuged at 2600× *g* (10 min, 6 °C). Next, the entire organic phase was transferred to a clean glass vial and evaporated under a nitrogen stream. The residue was then diluted in 4 mL acetonitrile, vortexed for 30 s, and 1 mL of distilled water was then added. The extract was transferred to a Captiva EMR-Lipid cartridge (Agilent, Santa Clara, CA, USA). The eluates were collected by gravity in a polypropylene centrifuge tube for the SALLE-DLLME procedure. Accordingly, 700 mg NaCl and 2 μL of formic acid (98%) were added and the solution was manually shaken for 60 s; then, after centrifugation at 2600× *g* for 10 min at 6 °C, the supernatant was placed in a 15 mL screw-cap glass test tube, concentrated to 1 mL under a nitrogen stream, and diluted with 10 mL of 10% NaCl aqueous solution (*w*/*v*) at pH of 2. Next, 1.5 mL TCM was injected, and the mixture was shaken for 60 s and centrifuged for 10 min at 2600× *g* (6 °C), transferring the whole sedimented phase into a chromatographic glass vial and then analyzed by gas chromatography/tandem mass spectrometry (GC–MS/MS).

The gas chromatography system consisted of an Agilent 7890 GC with split-splitless inlet and Agilent 7693 autosampler (Agilent, Santa Clara, CA, USA). HP–5MS-UI capillary column (30 m × 0.25 mm i.d.; 0.25 μm film thickness) from Agilent was used to separate target analytes. Samples were automatically injected in splitless mode, and the temperature of injection port and injection volume were 270 °C and 2 μL, respectively. The carrier gas with high purity (99.999%) was helium at constant flow (1.0 mL min^−1^). The initial oven temperature was set at 70 °C for 2.0 min, raised at a rate of 15 °C min^−1^ to 160 °C, held for 5 min, and finally, raised at rate of 30 °C min^−1^ to 300 °C, then held for 4 min (total time of 21.7 min).

The mass spectrometer was an Agilent 7000D triple quadrupole (Agilent, Santa Clara, CA, USA) equipped with inert electron-impact (EI, 70 ev) ion source and operated in dynamic multiple reaction monitoring (DMRM) mode. Two MS/MS transitions for each analyte were reported, the first for quantification and the second for confirmation. Temperatures of the transfer line, ion source, and quadrupoles were 280 °C, 280 °C, and 150 °C, respectively. The mass spectrometer was auto-tuned weekly. The limit of detection (LOD) was determined as the smallest amount of the analyte that gave a signal-to-noise ratio ≥ 3 and was set at 0.1 ng g^−1^ tissue for each POP.

### 4.3. RNA Isolation and Quantitative Real-Time Polymerase Chain Reaction (qRT-PCR)

Using QIAzol reagent (Qiagen, Hilden, Germany) and the RNeasy Mini kit (Qiagen, Hilden, Germany), total RNA was extracted from 30 mg of each endometriotic tissue sample following the manufacturer’s protocol, and the concentration was assessed through a NanoDrop 2000 spectrophotometer (Thermo Fisher Scientific, Waltham, MA, USA), verifying that the A260/A280 ratio remained within the range of 1.8 to 2.2 for each sample. An iScript cDNA synthesis kit (Bio-Rad Laboratories, Hercules, CA, USA) was used for reverse transcription according to the manufacturer’s instructions.

Real-time PCR was carried out with a CFX96 Real-time PCR detection system (Bio-Rad Laboratories, Hercules, CA, USA) using SsoAdvanced SYBR^®^ Green Supermix (Bio-Rad Laboratories, Hercules, CA, USA). Expression of 36 genes was measured according to the manufacturer’s protocol. All included genes are involved in the endometriosis-related EMT process. Particularly, 5 of them belongs to the core EMT markers [claudin 7 (CLDN7), integrin beta-2 (ITGB2), matrix metallopeptidase 1 (MMP1), matrix metallopeptidase 7 (MMP7), and tissue inhibitor metallopeptidase 1 (TIMP1)], 10 were related to the TGFβ and PDGFRα signaling pathways [anterior gradient protein 2 homolog (AGR2), dual specificity phosphatase 6 (DUSP6), forkhead box M1 (FOXM1), fucosyltransferase 8 (FUT8), high-mobility group AT-hook 1 (HMGA1), high-mobility group AT-hook 2 (HMGA2), homeobox A10 (HOXA10), platelet-derived growth factor receptor alpha (PDGFRA), POU class 5 homeobox 1 (POU5F1), and retinoid X receptor alpha (RXRA)], 4 related to the Wnt signaling pathway [proto-oncogene, polycomb ring finger (BMI1), cyclin-dependent kinase 1 (CDK1), dickkopf WNT signaling pathway inhibitor 1 (DKK1), and SRY-box transcription factor 2 (SOX2)], 4 to the Notch/NFκB-related signaling pathway [C-C motif chemokine receptor 1 (CCR1), component of inhibitor of nuclear factor kappa B kinase complex (CHUK), Fos proto-oncogene (FOS), and Jun proto-oncogene (JUN)], 5 were inflammation-related EMT inducers [interleukin 1 receptor, type II (IL1R2); interleukin 1 receptor, type I (IL1RL1); interleukin 6 cytokine family signal transducer (IL6ST); ribonucleotide reductase M2 (RRM2); and tumor necrosis factor receptor superfamily member 1B (TNFRSF1B)], 4 were estrogen-related EMT inducers [cytochrome P450 family 19 subfamily A member 1 (CYP19A1), epidermal growth factor (EGF), estrogen receptor 1 (ESR1), and fibroblast growth factor 2 (FGF2)], and 4 were hypoxia-related EMT inducers [angiogenin (ANG), angiopoietin 1 (ANGPT1), fms-related receptor tyrosine kinase 1 (FLT1), and vascular endothelial growth factor (VEGFA)]. The criteria for the selection of these genes were based on previous evidence relating exposure to EDCs with the expression of these genes and their participation in the endometriosis-related EMT process [40,44,47,80,99]. Primers used for these studies were purchased from Bio-Rad Laboratories (Hercules, CA, USA), and their assay ID, amplicon context sequence, amplicon length (bp), and chromosome location are detailed in Appendix A. Each mRNA primer employed in this investigation was meticulously crafted and validated by Bio-Rad Laboratories. The process involved devising primers, validating them through experimental assays using universal RNA, analyzing amplification plots and melt curves, and assessing efficiency and dynamic range through the generation of a seven-point standard curve. Subsequent to this, specificity was ascertained through next-generation sequencing of the resulting amplicon.

Gene expression was analyzed via Bio-Rad CFX 96 Manager Software 3.1 to determine the cycle of quantification (Ct), and this was calculated using the 2^−ΔΔCt^ method. All values were normalized using glyceraldehyde-3-phosphate dehydrogenase (GAPDH) RNA expression levels, calculating the difference corresponding to the quantification cycles between the target genes and GAPDH gene: ∆Ct = Ct (gene of interest) − Ct (GAPDH).

### 4.4. Statistical Analysis

Descriptive analysis was conducted of the sociodemographic, lifestyle, and gynecological characteristics of cases and controls. Continuous variables were expressed as means ± standard deviation, and categorical variables were expressed as percentages. Adipose tissue concentrations of individual OCP and PCB congeners, the sum of OCPs (∑OCPs), PCBs (∑PCBs), and POPs (∑POPs) were expressed as means ± standard deviation and percentiles (25, 50, and 75). The total molar concentration of OCPs (∑OCPs), PCBs (∑PCBs), and POPs (∑POPs) expressed in nmol/g tissue was also calculated as the sum of the molar concentrations of adipose tissue OCPs, PCBs, and POPs, respectively. Adipose concentrations of OCPs and PCBs below the LOD were assigned a value of LOD/2. Adipose concentrations of OCPs and PCBs were log-transformed to minimize the influence of extreme values.

Bivariate analyses were conducted between cases and controls using the chi-square and Student’s (or Mann–Whitney) tests as appropriate. The Mann–Whitney test was used to compare concentrations of POPs between cases and controls and between stage I/II cases and stage III/IV cases. Unconditional logistic regression analyses were performed to determine the odds ratios (ORs) for endometriosis risk of omental adipose concentrations of OCPs and PCBs in the study population. In additional models, concentrations of OCPs and PCBs were entered in terciles. Given the limited sample size, regression analyses were adjusted for (1) age (yr), (2) age (yr), and BMI (kg/m^2^) and (3) age (yr), BMI (kg/m^2^), residence (rural/urban), and parity (yes/no). Given the similarity in results obtained, only those from fully adjusted models are discussed. Results are summarized as ORs with corresponding 95% confidence intervals. Because the independent variables were log-transformed, OR estimates reflect the odds of endometriosis risk for each 1 log unit of the concentration of the corresponding biomarker.

Correlations between POP concentrations and gene expression levels were calculated by Spearman’s rank correlation coefficient for those genes expressed in >75% of the samples, while for those genes expressed in 25–75% of the samples, they were considered as dichotomous variables [according to the median value (for those between 50 and 75%) or to the minimum detectable level (for those between 25 and 50%)] and the Mann–Whitney test was applied. Associations with POPs were not explored for those genes expressed in <25% of the samples.

Given the small sample size, the significance level was set at *p* = 0.050, although results with *p*-values between 0.100 and 0.050 were also cautiously discussed. All tests were two-tailed, and SPSS Statistics 23.0 (IBM, Chicago, IL, USA) and R statistical computing environment v3.1 were used for data analyses, while figures were designed with GraphPad Prism 5.0 software (San Diego, CA, USA).

## Figures and Tables

**Figure 1 ijms-25-04420-f001:**
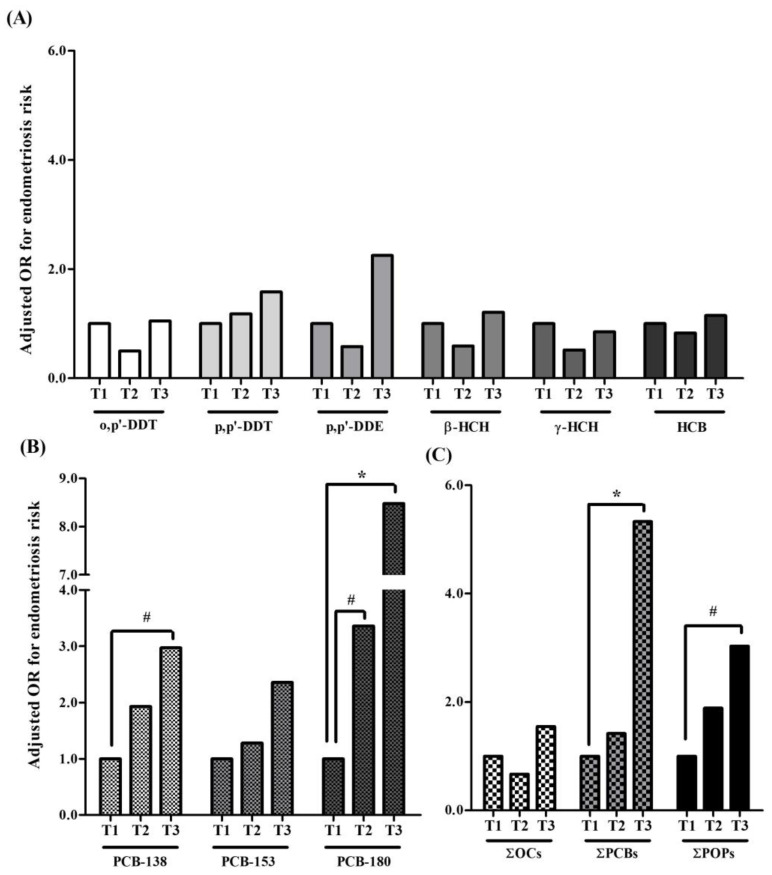
Adjusted odds ratios (OR) for endometriosis in the tertiles (T1, T2, T3) of omental adipose tissue concentrations of (**A**) organochlorine pesticides (OCs); (**B**) polychlorinated biphenyls (PCBs); and (**C**) the sum of OCs, PCBs, and POPs. Models were adjusted for age (yr), body mass index (kg/m^2^), residence (rural/urban), and parity (yes/no). # *p*-value 0.100–0.050 compared with T1; * *p*-value < 0.050 compared with T1.

**Table 1 ijms-25-04420-t001:** Omental adipose tissue concentrations of organochlorine pesticides and polychlorinated biphenyls (ng/g tissue).

		Cases (n = 34)					Controls (n = 75)					
	% >LOD	Mean	SD	Percentiles			Mean	SD	Percentiles			
				25th	50th	75th			25th	50th	75th	*p*-Value
*o,p’*-DDT	96.30	0.89	1.19	0.28	0.46	0.83	0.50	0.38	0.27	0.40	0.60	0.360
*p,p’*-DDT	89.90	3.50	4.94	0.94	1.72	3.61	1.84	1.84	0.19	1.40	2.53	0.065
*p,p’*-DDE	100.00	282.05	552.90	48.03	118.52	226.42	103.35	84.47	41.96	74.33	136.76	0.067
β-HCH	100.00	32.79	45.86	4.31	14.88	37.12	16.67	17.18	5.46	11.09	22.34	0.261
γ-HCH	100.00	25.10	34.27	5.07	11.27	28.91	12.80	12.53	4.60	8.32	16.80	0.204
HCB	100.00	15.43	14.81	5.05	6.88	24.57	9.99	7.32	5.26	7.62	11.68	0.513
∑OCPs ^1^	-	1.16	1.93	0.24	0.55	1.13	0.47	0.35	0.21	0.35	0.60	0.072
PCB-138	99.10	30.41	30.83	10.39	20.29	38.95	16.85	12.33	7.76	13.74	24.03	0.015
PCB-153	100.00	39.77	41.32	12.47	26.76	45.48	21.90	16.99	9.21	18.50	31.05	0.012
PCB-180	100.00	64.52	87.79	18.28	39.04	70.14	31.25	28.72	9.82	20.54	50.62	0.006
∑PCBs ^1^	-	0.36	0.42	0.11	0.25	0.37	0.19	0.15	0.07	0.15	0.27	0.009
∑POPs ^1^	-	1.51	2.15	0.43	0.76	1.78	0.66	0.46	0.30	0.52	0.92	0.018

^1^ Expressed in nmol/g tissue; LOD: limit of detection; SD: standard deviation; DDT: dichlorodiphenyltrichloroethane; DDE: dichlorodiphenyldichloroethylene; HCH: hexachlorohexane; HCB: hexachlorobenzene; OCP: organochlorine pesticides; PCB: polychlorinated biphenyls; POP: persistent organic pollutant.

**Table 2 ijms-25-04420-t002:** Relationship of omental adipose tissue concentrations of organochlorine pesticides and polychlorinated biphenyls with endometriosis. Logistic regression analyses (n = 109).

	OR	IC 95%		*p*-Value	OR ^1^	IC 95%		*p*-Value
*o,p’*-DDT	1.45	0.87	2.42	0.152	1.35	0.78	2.34	0.284
*p,p’*-DDT	1.62	1.14	2.32	0.008	1.63	1.13	2.34	0.008
*p,p’*-DDE	1.49	0.98	2.26	0.061	1.42	0.90	2.26	0.135
β-HCH	1.11	0.83	1.48	0.499	0.99	0.72	1.36	0.935
γ-HCH	1.25	0.84	1.86	0.268	1.08	0.69	1.67	0.747
HCB	1.26	0.75	2.13	0.382	1.16	0.63	2.10	0.636
∑OCPs ^1^	1.54	0.98	2.42	0.060	1.44	0.87	2.40	0.159
PCB-138	1.60	0.98	2.62	0.059	1.65	0.93	2.92	0.086
PCB-153	1.59	0.99	2.53	0.054	1.65	0.95	2.87	0.075
PCB-180	1.63	1.08	2.46	0.021	1.89	1.09	3.27	0.023
∑PCBs ^1^	1.64	1.04	2.58	0.034	1.78	1.02	3.12	0.044
∑POPs ^1^	1.76	1.07	2.88	0.025	1.75	0.99	3.10	0.056

^1^ Adjusted for age (yr), body mass index (kg/m^2^), residence (rural/urban), and parity (yes/no). OR: odds ratio; IC 95%: 95% confidence interval; DDT: dichlorodiphenyltrichloroethane; DDE: dichlorodiphenyldichloroethylene; HCH: hexachlorohexane; HCB: hexachlorobenzene; OCP: organochlorine pesticides; PCB: polychlorinated biphenyls; POP: persistent organic pollutant.

**Table 3 ijms-25-04420-t003:** Associations between endometriotic tissue EMT-related gene expression levels and omental adipose tissue concentrations of organochlorine pesticides and polychlorinated biphenyls (n = 30).

			*o’p’*-DDT	*p’,p’*-DDT	*p’p’*-DDE	β-HCH	γ-HCH	HCB	∑OCPs	PCB-138	PCB-153	PCB-180	∑PCBs	∑POPs
CORE EMT MARKERS	CLDN7 ^a^	ρ	**−0.46**	**−0.41**	**−0.49**	−0.26	−0.18	−0.29	**−0.45**	−0.18	−0.16	−0.16	−0.15	**−0.43**
		*p*-value	**0.011**	**0.026**	**0.005**	0.157	0.346	0.124	**0.013**	0.341	0.396	0.407	0.415	**0.018**
	ITGB2 ^a^	ρ	**0.59**	**0.52**	**0.56**	0.22	0.13	0.33	**0.49**	0.25	0.23	0.20	0.20	**0.49**
		*p*-value	**0.001**	**0.004**	**0.001**	0.247	0.493	0.072	**0.006**	0.176	0.223	0.300	0.289	**0.006**
	MMP1 ^b^	≤0.12	0.47 (0.53)	1.64 (3.44)	102.46 (171.71)	21.97 (64.91)	13.04 (46.69)	6.98 (22.06)	0.58 (0.80)	20.44 (31.95)	26.82 (41.64)	49.88 (52.72)	0.27 (0.34)	0.84 (1.27)
		>0.12	0.51 (0.85)	1.93 (2.69)	134.57 (369.65)	14.00 (28.98)	10.56 (20.85)	6.78 (11.77)	0.53 (1.36)	20.04 (27.62)	25.60 (30.15)	34.41 (39.29)	0.23 (0.21)	0.74 (1.82)
		*p*-value	0.885	0.548	0.468	0.520	0.373	0.787	0.694	0.663	0.604	0.310	0.373	0.983
	TIMP1 ^b^	≤0.39	0.58 (0.89)	1.43 (3.10)	102.46 (222.07)	14.66 (28.98)	10.97 (19.73)	6.44 (17.31)	0.48 (1.42)	19.16 (16.06)	25.41 (19.93)	34.41 (34.22)	0.22 (0.19)	0.68 (1.82)
		>0.39	0.47 (0.57)	1.93 (2.56)	134.57 (163.70)	19.34 (64.91)	13.41 (47.86)	7.87 (24.58)	0.58 (0.73)	22.82 (32.08)	30.94 (41.84)	51.76 (85.41)	0.28 (0.42)	0.77 (1.26)
		*p*-value	0.724	0.633	0.633	0.604	0.633	0.443	0.604	0.290	0.221	0.310	0.290	0.548
TGF-β/PDGFRα SIGNALING PATHWAYS	AGR2 ^b^	≤0.10	0.58 (1.91)	1.93 (5.68)	136.30 (220.21)	22.16 (59.93)	16.70 (46.20)	12.68 (21.97)	0.73 (1.69)	20.44 (33.71)	26.28 (42.24)	34.41 (55.54)	0.22 (0.35)	0.84 (1.83)
		>0.10	0.45 (0.38)	1.43 (1.75)	72.36 (183.63)	12.22 (23.46)	9.19 (15.64)	6.78 (5.33)	0.34 (1.22)	20.14 (9.93)	26.82 (8.86)	49.88 (47.68)	0.25 (0.12)	0.62 (1.43)
		*p*-value	0.237	0.191	0.395	0.237	0.191	0.548	0.351	0.724	0.633	0.310	0.548	0.694
	DUSP6 ^a^	ρ	−0.29	−0.31	−0.29	−0.31	−0.27	−0.23	−0.31	−0.20	−0.23	−0.13	−0.18	−0.29
		*p*-value	0.123	0.101	0.121	0.098	0.150	0.224	0.098	0.291	0.226	0.502	0.345	0.117
	FOXM1 ^a^	ρ	**0.61**	**0.54**	**0.53**	0.35	0.27	**0.39**	**0.50**	0.28	0.27	0.25	0.27	**0.51**
		*p*-value	**<0.001**	**0.002**	**0.003**	0.059	0.150	**0.034**	**0.005**	0.134	0.146	0.177	0.142	**0.004**
	FUT8 ^b^	≤0.09	0.41 (0.77)	1.41 (3.02)	75.28 (212.33)	14.33 (27.30)	10.76 (18.37)	6.50 (15.54)	0.33 (1.37)	19.04 (14.66)	25.50 (19.28)	34.82 (35.86)	0.23 (0.20)	0.54 (1.74)
		>0.09	0.57 (0.72)	2.50 (2.51)	171.75 (212.60)	25.05 (75.48)	18.88 (57.28)	10.83 (34.83)	0.79 (0.96)	33.05 (56.55)	42.33 (77.34)	60.48 (114.59)	0.36 (0.65)	1.11 (1.83)
		*p*-value	0.598	0.291	0.312	0.253	0.312	0.538	0.235	0.078	0.086	0.173	0.173	0.173
	HMGA1 ^a^	ρ	**0.57**	**0.50**	**0.56**	0.34	0.27	0.35	**0.51**	0.29	0.28	0.25	0.27	**0.53**
		*p*-value	**0.001**	**0.005**	**0.001**	0.067	0.152	0.061	**0.004**	0.119	0.134	0.183	0.156	**0.003**
	HOXA10 ^a^	ρ	0.32	0.27	**0.43**	0.20	0.17	0.13	**0.37**	0.20	0.18	0.16	0.20	**0.37**
		*p*-value	0.084	0.154	**0.018**	0.286	0.379	0.500	**0.043**	0.296	0.332	0.396	0.295	**0.047**
	PDGFRA ^a^	ρ	−0.15	−0.20	−0.08	−0.27	−0.26	−0.19	−0.17	−0.19	−0.20	−0.17	−0.19	−0.18
		*p*-value	0.433	0.291	0.686	0.148	0.173	0.303	0.358	0.316	0.295	0.373	0.307	0.331
	POU5F1 ^b^	≤0.04	0.45 (0.53)	1.33 (3.11)	72.36 (223.61)	9.54 (32.61)	6.01 (20.85)	6.44 (11.84)	0.27 (1.43)	20.04 (16.18)	25.6 (20.13)	35.24 (31.88)	0.23 (0.20)	0.50 (1.93)
		>0.04	0.51 (1.59)	2.42 (4.79)	136.3 (160.27)	21.97 (53.72)	13.41 (41.65)	8.98 (24.75)	0.73 (0.65)	22.82 (31.95)	30.94 (41.64)	51.76 (84.05)	0.28 (0.41)	0.84 (1.17)
		*p*-value	0.443	0.130	0.272	0.078	0.152	0.419	0.178	0.419	0.330	0.494	0.419	0.290
	RXRA ^a^	ρ	0.03	0.01	0.00	0.00	0.01	−0.03	−0.01	−0.06	−0.04	0.00	−0.02	−0.03
		*p*-value	0.857	0.961	0.987	0.983	0.948	0.886	0.964	0.754	0.843	0.995	0.931	0.886
WNT/β-CATENIN SIGNALING PATHWAY	BMI1 ^b^	≤0.58	0.51 (0.51)	1.83 (3.02)	153.84 (204.98)	19.34 (20.73)	13.04 (15.80)	8.98 (16.18)	0.58 (1.33)	20.14 (13.49)	26.82 (17.49)	40.76 (33.44)	0.25 (0.20)	0.84 (1.38)
		>0.58	0.47 (0.92)	1.81 (3.11)	78.19 (185.17)	15.11 (64.91)	11.39 (48.99)	6.44 (33.57)	0.36 (1.23)	20.44 (62.07)	26.70 (80.99)	37.32 (110.51)	0.25 (0.68)	0.58 (1.99)
		*p*-value	0.395	0.604	0.290	0.983	0.983	0.694	0.373	0.787	0.756	0.724	0.756	0.395
	DKK1 ^a^	ρ	**0.43**	**0.39**	**0.51**	0.22	0.14	0.11	**0.43**	0.09	0.07	−0.02	0.03	**0.40**
		*p*-value	**0.017**	**0.034**	**0.004**	0.243	0.469	0.572	**0.017**	0.644	0.714	0.899	0.872	**0.029**
	CDK1 ^a^	ρ	0.18	0.07	0.14	−0.08	−0.11	0.03	0.11	0.15	0.13	0.15	0.10	0.12
		*p*-value	0.355	0.709	0.466	0.673	0.546	0.856	0.546	0.434	0.505	0.431	0.607	0.511
	SOX2 ^b^	≤0.06	0.45 (0.53)	1.33 (2.77)	72.36 (222.07)	12.22 (32.49)	9.19 (20.70)	6.44 (11.84)	0.30 (1.42)	20.04 (13.49)	25.60 (17.49)	35.24 (31.88)	0.23 (0.19)	0.50 (1.82)
		>0.06	0.51 (1.59)	2.42 (4.79)	136.3 (163.70)	21.97 (57.03)	13.41 (43.55)	8.98 (24.75)	0.73 (0.65)	22.82 (35.82)	30.94 (45.47)	51.76 (85.41)	0.28 (0.44)	0.84 (1.17)
		*p*-value	0.576	0.191	0.494	0.206	0.351	0.468	0.351	0.633	0.520	0.694	0.633	0.419
NOTCH/NF-κB SIGNALING PATHWAY	CCR1 ^a^	ρ	**0.60**	**0.56**	**0.56**	0.33	0.24	0.27	**0.51**	0.18	0.18	0.06	0.13	**0.48**
		*p*-value	**<0.001**	**0.001**	**0.001**	0.070	0.206	0.154	**0.004**	0.332	0.349	0.765	0.492	**0.008**
	CHUK ^a^	ρ	0.13	0.05	0.14	−0.16	−0.19	−0.07	0.06	−0.01	−0.04	−0.03	−0.07	0.06
		*p*-value	0.505	0.793	0.445	0.393	0.327	0.718	0.741	0.938	0.825	0.890	0.732	0.761
	FOS ^b^	≤0.10	0.58 (0.78)	1.64 (2.91)	134.57 (175.97)	19.34 (58.01)	13.04 (42.16)	6.78 (21.76)	0.53 (0.98)	20.44 (30.51)	26.82 (39.83)	40.76 (58.48)	0.25 (0.36)	0.74 (1.55)
		>0.10	0.47 (0.89)	1.83 (3.38)	98.56 (433.72)	15.11 (26.53)	11.39 (18.25)	6.98 (8.98)	0.58 (1.95)	20.04 (15.97)	26.70 (21.60)	35.24 (58.33)	0.25 (0.23)	0.77 (2.13)
		*p*-value	0.700	0.910	0.700	0.803	0.769	0.455	0.734	0.512	0.455	0.769	0.603	0.734
	JUN ^b^	≤0.10	0.58 (1.13)	2.12 (3.75)	118.52 (260.86)	17.23 (54.56)	12.21 (39.36)	8.42 (18.39)	0.50 (1.49)	20.28 (28.56)	27.43 (33.01)	34.82 (55.10)	0.24 (0.27)	0.71 (1.90)
		>0.10	0.42 (0.30)	1.60 (1.33)	126.2 (176.58)	15.76 (24.21)	11.84 (16.70)	5.05 (17.87)	0.58 (0.67)	20.29 (46.32)	26.76 (59.60)	50.92 (84.81)	0.26 (0.50)	0.81 (1.02)
		*p*-value	0.482	0.542	0.778	0.963	0.888	0.223	0.851	0.851	0.888	0.373	0.606	0.888
INFLAMMATION SIGNALING PATHWAY	IL1RL1 ^a^	ρ	−0.01	−0.06	0.02	−0.13	−0.14	−0.26	−0.03	−0.14	−0.14	0.01	−0.08	−0.01
		*p*-value	0.950	0.759	0.917	0.496	0.467	0.173	0.859	0.477	0.456	0.964	0.675	0.965
	IL6ST ^a^	ρ	0.25	0.14	0.24	−0.14	−0.18	−0.02	0.14	0.02	−0.01	−0.04	−0.04	0.14
		*p*-value	0.186	0.462	0.200	0.469	0.341	0.927	0.469	0.919	0.960	0.827	0.827	0.468
	RRM2 ^a^	ρ	**0.58**	**0.50**	**0.46**	0.32	0.21	**0.43**	**0.43**	0.26	0.28	0.30	0.29	**0.46**
		*p*-value	**0.001**	**0.005**	**0.012**	0.088	0.263	**0.019**	**0.018**	0.173	0.139	0.107	0.123	**0.010**
	TNFRSF1B ^b^	≤0.10	0.36 (0.49)	**1.38 (2.43)**	64.58 (203.62)	14.00 (27.98)	10.56 (18.17)	6.56 (15.30)	0.30 (1.33)	20.04 (19.78)	26.28 (23.82)	37.32 (39.44)	0.25 (0.20)	0.58 (1.69)
		>0.10	0.63 (1.58)	**2.88 (4.66)**	164.28 (262.20)	23.82 (86.83)	17.93 (65.67)	12.68 (27.80)	0.74 (1.33)	24.04 (56.94)	30.94 (82.43)	51.96 (121.48)	0.27 (0.69)	0.85 (2.43)
		*p*-value	0.081	**0.032**	0.081	0.108	0.167	0.541	0.067	0.402	0.428	0.635	0.667	0.118
ESTROGEN/GROWTH FACTOR SIGNALING PATHWAY	EGF ^b^	≤0.10	0.63 (1.92)	2.57 (5.65)	153.84 (204.98)	21.97 (45.83)	13.41 (36.52)	10.24 (18.15)	0.74 (1.63)	20.14 (29.38)	26.82 (39.20)	40.76 (66.48)	0.26 (0.39)	0.84 (1.75)
		>0.10	0.47 (0.28)	1.81 (2.08)	78.19 (183.63)	14.66 (32.33)	10.97 (20.70)	6.44 (12.59)	0.36 (1.22)	20.44 (19.16)	26.70 (24.00)	37.32 (56.53)	0.25 (0.26)	0.58 (1.47)
		*p*-value	0.395	0.272	0.330	0.419	0.548	0.206	0.272	0.443	0.443	0.520	0.443	0.373
	ESR1 ^a^	ρ	**0.37**	0.33	**0.43**	0.12	0.05	0.09	0.35	0.05	0.03	0.00	0.02	0.33
		*p*-value	**0.044**	0.075	**0.018**	0.513	0.791	0.627	0.060	0.775	0.855	0.985	0.902	0.078
	FGF2 ^b^	≤0.12	0.58 (1.97)	1.43 (5.73)	102.46 (222.07)	14.00 (23.46)	10.56 (15.64)	6.44 (18.76)	0.48 (1.73)	18.92 (25.94)	25.41 (29.95)	34.41 (31.36)	0.22 (0.20)	0.68 (1.82)
		>0.12	0.47 (0.55)	1.93 (2.20)	134.57 (183.09)	19.34 (59.93)	13.41 (46.20)	7.87 (21.90)	0.58 (1.14)	22.82 (32.08)	30.94 (41.84)	51.96 (85.98)	0.28 (0.42)	0.77 (1.39)
		*p*-value	0.950	0.820	0.852	0.395	0.351	0.663	0.694	0.310	0.272	0.310	0.351	0.576
HYPOXIA SIGNALING PATHWAY	ANG ^a^	ρ	**0.41**	**0.37**	**0.37**	0.20	0.14	0.23	0.31	0.17	0.16	−0.01	0.05	0.30
		*p*-value	**0.024**	**0.047**	**0.043**	0.290	0.471	0.228	0.094	0.357	0.398	0.940	0.775	0.113
	ANGPT1 ^a^	ρ	−0.03	−0.11	−0.05	−0.26	−0.27	−0.28	−0.13	−0.25	−0.28	−0.19	−0.24	−0.14
		*p*-value	0.883	0.549	0.783	0.166	0.145	0.130	0.506	0.178	0.131	0.302	0.202	0.472
	FLT1 ^b^	≤0.35	0.36 (2.00)	1.38 (6.07)	64.58 (223.61)	14.66 (52.56)	10.97 (38.61)	5.79 (19.19)	0.30 (1.74)	18.48 (28.02)	25.41 (31.74)	34.41 (37.37)	0.22 (0.21)	0.58 (1.99)
		>0.35	0.51 (0.51)	2.42 (1.97)	136.30 (158.76)	19.34 (25.46)	13.41 (19.73)	8.98 (21.59)	0.73 (1.09)	22.82 (23.45)	30.94 (30.57)	51.76 (73.53)	0.27 (0.31)	0.85 (1.38)
		*p*-value	0.419	0.330	0.272	0.633	0.604	0.237	0.272	0.141	0.191	0.237	0.290	0.206
	VEGFA ^b^	≤0.92	0.47 (0.60)	1.81 (3.44)	78.19 (222.07)	14.66 (32.33)	10.97 (20.70)	6.44 (12.59)	0.36 (1.42)	20.44 (19.16)	26.28 (24.00)	35.24 (64.59)	0.22 (0.26)	0.62 (1.47)
		>0.92	0.58 (0.85)	1.83 (2.69)	136.30 (166.54)	22.16 (45.93)	16.70 (36.52)	10.24 (24.05)	0.58 (1.14)	20.14 (23.58)	26.82 (29.39)	48.07 (37.71)	0.27 (0.24)	0.85 (1.75)
		*p*-value	0.443	0.494	0.548	0.443	0.576	0.120	0.468	0.419	0.443	0.310	0.237	0.395

^a^ Expressed as Spearman rho coefficients (ρ) and *p*-values; ^b^ expressed as median [interquartilic range] and *p*-value: EMT: epithelial–mesenchymal transition; CLDN7: claudin 7; ITGB2: integrin beta-2; MMP1: matrix metallopeptidase 1; TIMP1: tissue inhibitor metallopeptidase 1; AGR2: Anterior gradient protein 2 homolog; DUSP6: dual specificity phosphatase 6; FOXM1:forkhead box M1; FUT8: fucosyltransferase 8; HMGA1: high mobility group AT-hook 1; HOXA10: Homeobox A10; PDGFRA: platelet-derived growth factor receptor alpha; POU5F1: POU class 5 homeobox 1; RXRA: retinoid X receptor alpha; BMI1 proto-oncogene, polycomb ring finger; CDK1: cyclin-dependent kinase 1; DKK1: dickkopf WNT signaling pathway inhibitor 1; SOX2: SRY-box transcription factor 2; CCR1: C-C motif chemokine receptor 1; CHUK: component of inhibitor of nuclear factor kappa B kinase complex; FOS: Fos proto-oncogene; JUN: Jun proto-oncogene; IL1RL1: interleukin 1 receptor, type I; IL6ST: interleukin 6 cytokine family signal transducer; RRM2: ribonucleotide reductase M2; TNFRSF1B: tumor necrosis factor receptor superfamily member 1B; EGF: epidermal growth factor; ESR1: estrogen receptor 1; FGF2: fibroblast growth factor 2; ANG: angiogenin; ANGPT1: angiopoietin 1; FLT1: fms related receptor tyrosine kinase 1; VEGFA: vascular endothelial growth factor A.

**Table 4 ijms-25-04420-t004:** Sociodemographic, anthropometric, and reproductive characteristics of endometriosis cases and controls.

	Cases (n = 34)	Controls (n = 75)	*p*-Value
	n (%)	n (%)	
**Age (years) ^1^**	35.9 ± 5.3	35.1 ± 8.2	0.967
**Height (m) ^1^**	1.6 ± 0.1	1.6 ± 0.1	0.365
**Weight (kg) ^1^**	69.5 ± 15.7	65.7 ± 13.1	0.600
**Body mass index (kg/m^2^) ^1^**	25.9 ± 5.5	24.6 ± 4.8	0.288
Normal weight (BMI < 25)	18 (52.9)	46 (61.3)	0.166
Overweight (BMI 25–30)	7 (20.6)	20 (26.7)	
Obese (BMI > 30)	9 (26.5)	9 (12.0)	
**Residence**			0.152
Rural	10 (29.4)	13 (17.3)	
Urban	24 (70.6)	62 (82.7)	
**Educational level**			0.949
Less than university degree	22 (64.7)	49 (65.3)	
University degree	12 (35.3)	26 (34.7)	
**Working outside home**			0.766
Yes	24 (70.6)	55 (73.3)	
No	10 (29.4)	20 (26.7)	
**Current Smoker**			0.549
Yes	24 (70.6)	57 (76.0)	
No	10 (29.4)	18 (24.0)	
**Parity**			0.593
Nulliparous	14 (41.2)	35 (46.7)	
Primiparous/multiparous	20 (58.8)	40 (53.3)	
**Intensity of menstrual bleeding**			0.542
Mild	11 (32.4)	20 (26.7)	
Moderate/Severe	23 (67.6)	55 (73.3)	
**rASRM endometriosis classification**			-
I	10 (29.4)	-	
II	12 (35.3)	-	
III	8 (23.5)	-	
IV	4 (11.8)	-	

^1^ Mean ± standard deviation.

## Data Availability

Data are contained within the article and Appendix A.

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
