# Peer review of "Environmental Exposure to Persistent Organic Pollutants and Its Association with Endometriosis Risk: Implications in the Epithelial–Mesenchymal Transition Process"

_ijms, 2024, doi:10.3390/ijms25084420_

Round 1

Reviewer 1 Report

Comments and Suggestions for Authors

The present study was investigated that the relationship of adipose tissue concentrations of some persistent organic pollutants (POPs) with the risk of endometriosis, and the endometriotic tissue expression profile of genes related to the endometriosis-related epithelial-mesenchymal transition (EMT) process. As a results, authors suggested that exposure to POPs may increase the risk of endometriosis and might have a role on the endometriosis related EMT development, contributing to the disease onset and progression. Further studies are warranted to corroborate these findings.

This manuscript is written well; however this manuscript will need to revise a few points.

1) Authors described that given the current discrepancies on the influence of POPs in the development of endometriosis, the present study was designed to explore the association between the bioaccumulated concentrations of OCs and PCBs, and the risk of endometriosis in women of childbearing age. Finally, we assessed the influence of this exposure on the expression profiles of EMT-related genes in the endometriotic tissue. So, could you show this study's hypothesis more exactly?

2)The choice of target examples is not clear. Please explain in detail using diagrams.

3)Regarding the figures, please make them a little more understandable and the resolution clearer. Many readers may not understand this figure. If possible, could you use collars in this figure?

4)Please revisit your results-based considerations. Please use the cited references to support the results of this study.

5)Authors showed that the relatively small sample size, reducing the statistical power and the possibility to examine differences in risk between stages of endometriosis and associations with gene expression profiles in endometriotic tissues etcAre there any other limitations to the study? Please be more specific.

Other

Please check again how to cite the literature. Does this paper meet the submission requirements?

Comments on the Quality of English Language

N/A

Author Response

The present study was investigated that the relationship of adipose tissue concentrations of some persistent organic pollutants (POPs) with the risk of endometriosis, and the endometriotic tissue expression profile of genes related to the endometriosis-related epithelial-mesenchymal transition (EMT) process. As a results, authors suggested that exposure to POPs may increase the risk of endometriosis and might have a role on the endometriosis related EMT development, contributing to the disease onset and progression. Further studies are warranted to corroborate these findings.

This manuscript is written well; however, this manuscript will need to revise a few points.

Response: We are grateful for the positive evaluation by this reviewer, whose concerns have been addressed in the revised manuscript, as detailed below.

1) Authors described that given the current discrepancies on the influence of POPs in the development of endometriosis, the present study was designed to explore the association between the bioaccumulated concentrations of OCs and PCBs, and the risk of endometriosis in women of childbearing age. Finally, we assessed the influence of this exposure on the expression profiles of EMT-related genes in the endometriotic tissue. So, could you show this study's hypothesis more exactly?

Response: Following reviewer comment, we have rewritten the hypothesis. The new version of the manuscript now reads:

“Thus, given the current discrepancies on the influence of POPs in the development of endometriosis, the present study was designed to explore the association between the bioaccumulated concentrations of OCs and PCBs, and the risk of endometriosis in women of childbearing age. Finally, given the crucial role of the EMT signaling pathway in dis-ease onset and the in vitro evidence of the link between POP exposure and EMT, we also assessed the influence of this exposure on the expression profiles of EMT-related genes in the endometriotic tissue.”

2)The choice of target examples is not clear. Please explain in detail using diagrams.

Response: As suggested by reviewer, a diagram has been created to facilitate readership understanding on the specific role of each selected gene on the EMT signaling pathway. This has been mentioned in the new version of the manuscript, that now reads:

“As schematized in Supplementary Figure 1, EMT is a physiological process in which cells lose the epithelial features such as E-cadherin, desmoplakin, occludings and claudins, and gain properties of mesenchymal cells such as N-cadherin, vimentin, and some integ-rin heterodimers.”

3)Regarding the figures, please make them a little more understandable and the resolution clearer. Many readers may not understand this figure. If possible, could you use collars in this figure?

Response: Following the reviewer suggestion, we have modified Figure 1 by grouping associations according to chemicals subfamilies: organochlorine pesticides (Figure 1A), PCBs (Figure 1B), and the sum of organochlorine pesticides, PCBs and POPs (Figure 1C). In order to facilitate readership comprehension, we have also rotated the graphs, showing the tertiles of exposure in the X-axis and the aORs in the Y-axis. Moreover, we have included brackets to facilitate readers’ understanding of the observed significant (and close-to-significant) associations.

4)Please revisit your results-based considerations. Please use the cited references to support the results of this study.

Response: We are not sure about what the reviewer means with this comment. As can be seen below, along the discussion section, we have compared point by point our results with those published by other researchers previously:

“In this study, we have observed that the CLDN7 epithelial marker in the endometriotic lesions was inversely associated with some POPs, while the ITGB2 mesenchymal marker was positively associated with them, suggesting that POP exposure might favors EMT in endometriotic lesions. In this regard, Zucchini-Pascal, et al. [65] demonstrated that in vitro exposure to three OCPs caused the repression of epithelial markers (cell-cell junctions and E-cadherin) and increased abundance of various mesenchymal genes, including vimentin, fibronectin, and its receptor ITGA5 in human primary cultured hepatocytes.  Bratton, et al. [80] also reported up-regulation of ITGA6 in breast cancer cells after expo-sure to DDT, another marker of mesenchymal phenotype [81]. Moreover, Hu, et al. [69] observed that exposure to PCB-104 of primary cultures of eutopic endometrial tissues from women with endometriosis increased migration and invasion through the up-regulation of mesenchymal markers such as MMP3 and MMP10 gene expression. Similarly, endometrial stromal cells exposed to HCB also enhanced migration and invasiveness ability involving ER signaling pathway [82].”

“These results are in line with previous studies suggesting upregulation of PDGFRα gene expression in estrogen-dependent cells such as epithelial mammary cells exposed to a pesticide mixture [83]. DDT has also been shown to activate p38 MAPK pathway [84], one of the main intracellular cascades driven by TGFβ/PDGFRα receptors in human endometrial adenocarcinoma cells. Human hepatocyte cultures exposed to some OCPs also increased the expression of DKK1 gene [85]. It has also been recently reported that in vivo models of endometrial cancer exposed to other endocrine disrupting chemical such as diethylstilbestrol (DES) aberrantly activated the WNT/β-catenin and TGFβ/PDGFRα-related PI3K/AKT signaling pathways [86].”

“…which is supported by previous evidence that also reported up-regulation of IL6R, TNF and CXCL8 genes after exposure to DDT in breast cancer cells [80,91]. Human hepatocyte cultures exposed to some OCPs also upregulated pro-inflammatory markers such as IL17RB [85]. Moreover, in line with previous studies on mouse thymoma cells exposed to tributyltin oxide, another EDC [92], we have observed positive correlations between some POPs and increased expression of RRM2 gene, which has been reported to promote EMT through the STAT3 cascade [63].”

“In line with these findings, Grünfeld and Bonefeld-Jorgensen [20] and Bratton, et al. [80] also reported up-regulation of hormone receptors such as ESR1 and progesterone receptor (PGR) in breast cancer cells after exposure to DDT and other OCPs. Regarding hypoxia-related genes, VEGF and other pro-angiogenic factors, such as ANG and ANGPT1, are markedly expressed in hypoxic conditions and suspected to promote EMT though the NFκB and β-catenin intracellular cascades [94]. We have observed that ANG gene expression was positively correlated to concentrations of o,p’-DDT, p,p’-DDT and p,p’-DDE concentrations, in line with numerous studies that have reported OCP-related up-regulation of pro-angiogenic gene mRNA levels [80,95,96].  ”

In all these paragraphs, we have explained the associations observed by other researchers and if they are in line or not with our findings. As reviewer may know, there is a scarcity of epidemiological evidence on the relationship between human exposure to POPs and gene expression disturbances in endometriotic tissue. For that reason, we have mainly compared our findings with the relatively low number of in vitro and in vivo studies addressing POP exposure and EMT-gene expression perturbation in a variety of cell lines/primary cultures and mouse models. In this regard, we have included in the discussion, to our knowledge, all the studies exploring this topic. In case that the reviewer has identified any article not included in the discussion of this work, we would appreciate it if they could inform us so that we may include it.

5)Authors showed that the relatively small sample size, reducing the statistical power and the possibility to examine differences in risk between stages of endometriosis and associations with gene expression profiles in endometriotic tissues etc. Are there any other limitations to the study? Please be more specific.

Response: We are grateful for this comment. We have revised this paragraph of the discussion section, and it has been rewritten. As shown in the revised version of the manuscript, in addition to the reduced sample size, we have clearly stated other limitations of this study, including (i) the inability to compare the results obtained in endometriotic tissue with healthy endometrium, (ii) the constrained availability of endometriotic tissue samples that hindered our ability to conduct protein validation analysis, thereby confirming the reported gene expression outcomes, (iii) the variety of endometriotic samples included that may reflect different gene expression profiles, (iv) the exploration of the influence of individualized chemicals instead of mixtures, and (v) the limitation regarding the study design, that precluded us from making inferences regarding a causal relationship between POPs and both the risk of endometriosis and the profiles of gene expression:

“Additionally, given that endometrium samples were not collected, the observed correlations between exposure and EMT gene expression in endometriotic tissue was not compared to those in endometrium.”

“Moreover, the limited amount of endometriotic tissue sample available prevented us from performing protein validation analysis to confirm the reported gene expression results.”

“Furthermore, we did not considered in this study the location from the analyzed endometriotic samples (endometriomas, deep infiltrating lesions,…) that might show different gene expression profiles.”

“In addition, we cannot rule out that the associations found with single chemicals are a surrogate of exposure to other unmeasured pollutants with similar physicochemical proper-ties or even to mixtures of pollutants that exert a combined effect.”

“Finally, our case-control/cross-sectional design prevented us from inferring a causal association between POPs and both endometriosis risk and gene expression profiles, although adipose tissue is the preferential reservoir for POP accumulation, and concentrations in this matrix are considered indicative of long-term exposure [97].”

Other: Please check again how to cite the literature. Does this paper meet the submission requirements?

Response: We have used a reference manager, and the document has been adapted to MDPI requirements. In case that any typographical error has been identified, we would correct it.

Reviewer 2 Report

Comments and Suggestions for Authors

1. Title is well chosen and reflects the topic of the study.

2. Abstract - no issues detected

3. introduction : it is well written and pleasant to read

v52-55  - the sentence suggest that the endometriosis causes autoimmunological disorders, but it is correlation only. we can not say if there is a cause-effect relationship and in which direction. the sentence has to be revised.

4. Results:

"Characteristics of the study population" - in reviewer opinion the section should be moved to "material and methods".

among cases there are 2 times more women with obesity than in control women - in my opinion the statistics regarding BMI/body weight is to simple and further details should be presented - mean BMI, mean weight. In a present form I do not believe that the groups are similar and it could be a weakness of study that should be discussed and indicated.

  5. Material and methods.

It is not clear what type of endometrial /endometriotic tissue was analysed? endometrium form cavity? endometriotic lesions? what type of lesions? 

If tissue was collected form endometriotic lesions how to compare them to controls that not have endometriotic lesions?

Why the endometrium was not analysed?

Author Response

REVIEWER 2

We are grateful for the positive evaluation by this reviewer, whose concerns have been addressed in the revised manuscript, as detailed below.

  1. Title is well chosen and reflects the topic of the study.

Response: We appreciate the favorable evaluation provided by this reviewer.

  1. Abstract - no issues detected

Response: We express our gratitude for the favorable evaluation provided by this reviewer.

  1. introduction : it is well written and pleasant to read

v52-55 - the sentence suggest that the endometriosis causes autoimmunological disorders, but it is correlation only. we can not say if there is a cause-effect relationship and in which direction. the sentence has to be revised.

Response: We are grateful for this constructive comment. As suggested, we have modified the sentence, that now reads:

“Furthermore, it has been observed that endometriosis correlates with increased risk for autoimmune disorders, fibromyalgia, chronic fatigue syndrome, gynecological cancers, and adenomyosis in affected women [8-11].”

  1. Results:

"Characteristics of the study population" - in reviewer opinion the section should be moved to "material and methods".

Response: Although we are grateful for the suggestion of this reviewer, we strongly consider important to maintain this section in the result section, given that readership must be conscious that women included in the case and the control groups were comparable in terms of sociodemographic, lifestyle and clinical characteristics. Moreover, taking into consideration that IJMS format requirements place the material and methods section after the result section, we consider of relevance to state this comparability between cases and controls before the main results of the manuscript. Nevertheless, in case that editor agree with the reviewer, we would move this paragraph to the material and methods section. 

among cases there are 2 times more women with obesity than in control women - in my opinion the statistics regarding BMI/body weight is to simple and further details should be presented - mean BMI, mean weight. In a present form I do not believe that the groups are similar and it could be a weakness of study that should be discussed and indicated.

Response: We agree with the reviewer that the percentage of obese women in the case group doubled the percentage in the control group. However, as reviewer can see in Table 1, the differences in the percentage of women in each category was not significant (p-value = 0.166). Moreover, in addition to the percentage of women in each BMI category, we clearly shown the mean ± standard deviation of the BMI in the case group (25.9±5.5 Kg/m2) and the control group (24.6±4.8 Kg/m2), with no significant differences between groups (p-value = 0.288). Similarly, in Table 1 we have also included the mean (± standard deviation) height and weight of women from both groups. As shown, no significant differences were observed in these two variables either (p-values = 0.365 and 0.600, respectively).

Nevertheless, the unconditional logistic regression analyses performed to determine the odds ratios (ORs) for endometriosis risk of omental adipose concentrations of OCPs and PCBs in the study population were adjusted for BMI (among other covariates), in order to exclude any influence of this variable in the associations found.

Taken all this information together, we strongly consider that main findings of this study are robust and not influenced by the BMI of women, given that the BMI of cases and controls are comparable and even in the scenario of any differences in terms of this variable, this was removed in the main analysis, as it was included as covariate.

  1. Material and methods.

It is not clear what type of endometrial /endometriotic tissue was analysed? endometrium form cavity? endometriotic lesions? what type of lesions?

Response: As can be seen in section 4.1, we collected endometriotic tissue, namely samples from the endometriotic lesions. Given the variety on the lesion location between women, there was not uniformity in the samples analyzed. This information has been added to the revised version of the manuscript, that now reads:

“Furthermore, we did not considered in this study the location from the analyzed endometriotic samples (endometriomas, deep infiltrating lesions,…) that might show different gene expression profiles.”

If tissue was collected form endometriotic lesions how to compare them to controls that not have endometriotic lesions?

Response: We are grateful for this constructive comment. We agree with the reviewer that the information regarding the study design was confusing, given that we stated that we carried out a case-control study, but we did not mention the cross-sectional design carried out for the secondary objective of this study. Now, we have clearly stated that we combined two study designs: we conducted (i) a case-control study for the main objective (to explore the association between POP exposure and the risk of endometriosis) and (ii) a cross-sectional design for the secondary objective (to assess the influence of POP exposure on the expression profiles of EMT-related genes in the endometriotic tissue). This information has been now included in the revised version of the manuscript that now reads:

Introduction section:

“…and EMT, the secondary aim of this study was to assess the influence of this exposure on the expression profiles of EMT-related genes in the endometriotic tissue.”

Material and method section:

“For the secondary aim of this study, a cross-sectional design was adopted, including only women from the case group.”

Moreover, as reviewer noted, we did not compare endometriotic gene expression profiles with those in healthy tissues from controls. Given that this could be a limitation of this study, this information was added to the revised version of the manuscript, that now reads:

“Additionally, given that endometrium samples were not collected, the observed correlations between exposure and EMT gene expression in endometriotic tissue was not compared to those in endometrium.”

Why the endometrium was not analysed?

Response: As mentioned above, endometrium samples were not collected for this research project. Despite now strongly considering the valuable information that endometrium samples would have added to this study, during its conceptualization we decided not to include this biological sample given the complexity of the whole project (we collected urine, blood, omental adipose tissue and endometriotic tissue simultaneously). Nevertheless, as this issue is an important limitation of the study, we have included this information in the discussion section, that now reads:

“Additionally, given that endometrium samples were not collected, the observed correlations between exposure and EMT gene expression in endometriotic tissue was not compared to those in endometrium.”

Round 2

Reviewer 2 Report

Comments and Suggestions for Authors

Authors revised manuscript and modified it according to the suggestions.

Last issue:

"Characteristics of the study population" - in reviewer opinion the section should be moved to "material and methods".

Response: Although we are grateful for the suggestion of this reviewer, we strongly consider important to maintain this section in the result section, given that readership must be conscious that women included in the case and the control groups were comparable in terms of sociodemographic, lifestyle and clinical characteristics. Moreover, taking into consideration that IJMS format requirements place the material and methods section after the result section, we consider of relevance to state this comparability between cases and controls before the main results of the manuscript. Nevertheless, in case that editor agree with the reviewer, we would move this paragraph to the material and methods section. 

The order of sections is not important. Many journals put material and method on the end. You can read it in any sequence as you wish. It is nice that you have well recruited  study and control groups - if not the study will not be scientific at all - but this is not the result of study but the result of inclusion process. The most important issue what is to distinguish between "material"-what you put into the study and "results" - outcomes of your research. 

Author Response

Authors revised manuscript and modified it according to the suggestions.

Response: We are very grateful for the positive evaluation

Last issue:

"Characteristics of the study population" - in reviewer opinion the section should be moved to "material and methods".

Authors’ Response: Although we are grateful for the suggestion of this reviewer, we strongly consider important to maintain this section in the result section, given that readership must be conscious that women included in the case and the control groups were comparable in terms of sociodemographic, lifestyle and clinical characteristics. Moreover, taking into consideration that IJMS format requirements place the material and methods section after the result section, we consider of relevance to state this comparability between cases and controls before the main results of the manuscript. Nevertheless, in case that editor agree with the reviewer, we would move this paragraph to the material and methods section. 

Reviewer’s Response: The order of sections is not important. Many journals put material and method on the end. You can read it in any sequence as you wish. It is nice that you have well recruited  study and control groups - if not the study will not be scientific at all - but this is not the result of study but the result of inclusion process. The most important issue what is to distinguish between "material"-what you put into the study and "results" - outcomes of your research. 

Response: According to reviewer suggestion, this part of the manuscript has been moved to the material and method section.
